# Anti-Cancer Effects of α-Cubebenoate Derived from *Schisandra chinensis* in CT26 Colon Cancer Cells

**DOI:** 10.3390/molecules27030737

**Published:** 2022-01-23

**Authors:** Jeong Eun Gong, Ji Eun Kim, Su Jin Lee, Yun Ju Choi, You Jeong Jin, Young Whan Choi, Sun Il Choi, Dae Youn Hwang

**Affiliations:** 1Laboratory Animal Resources Center, Department of Biomaterials Science (BK21 FOUR Program), College of Natural Resources & Life Science, Pusan National University, Miryang 50463, Korea; jegog@naver.com (J.E.G.); prettyjiunx@naver.com (J.E.K.); nuit4510@naver.com (S.J.L.); poiu335@naver.com (Y.J.C.); hjinyuu1@naver.com (Y.J.J.); 2Department of Horticultural Bioscience, Pusan National University, Miryang 50463, Korea; ywchoi@pusan.ac.kr; 3School of Pharmacy, Henan University, Kaifeng 475004, China; sunil.choi@hotmail.com; 4Longevity & Wellbeing Research Center and Laboratory Animals Resources Center, College of Natural Resources and Life Science, Pusan National University, Miryang 50463, Korea

**Keywords:** α-cubebenoate, CT26 colon cancer, apoptosis, migration, FAK/MLC

## Abstract

α-Cubebenoate derived from *Schisandra chinensis* has been reported to possess anti-allergic, anti-obesity, and anti-inflammatory effects and to exhibit anti-septic activity, but its anti-cancer effects have not been investigated. To examine the anti-cancer activity of α-cubebenoate, we investigated its effects on the proliferation, apoptosis, and metastasis of CT26 cells. The viabilities of CT26 cells (a murine colorectal carcinoma cell line) and HCT116 cells (a human colon cancer cell line) were remarkably and dose-dependently diminished by α-cubebenoate, whereas the viability of CCD-18Co cells (a normal human fibroblast cell line) were unaffected. Furthermore, α-cubebenoate treatment increased the number of apoptotic CT26 cells as compared with Vehicle-treated cells and increased Bax, Bcl-2, Cas-3, and Cleaved Cas-3 protein levels by activating the MAP kinase signaling pathway. α-Cubebenoate also suppressed CT26 migration by regulating the PI3K/AKT signaling pathway. Furthermore, similar reductions were observed in the expression levels of some migration-related proteins including VEGFA, MMP2, and MMP9. Furthermore, reduced VEGFA expression was found to be accompanied by the phosphorylations of FAK and MLC in the downstream signaling pathway of adhesion protein. The results of the present study provide novel evidence that α-cubebenoate can stimulate apoptosis and inhibit metastasis by regulating the MAPK, PI3K/AKT, and FAK/MLC signaling pathways.

## 1. Introduction

Natural products derived from microbes, plants, and marine organisms are considered major resources of potential chemotherapeutic agents [1]. The chiral centers, complex ring systems, and certain heteroatoms of these compounds cause the enhancement of the resultant biological diversity [2]. The cytotoxicity of cancer chemotherapeutics derived from natural products is mediated by various mechanisms that include the suppression of cell growth, the stimulation of apoptosis, and the inhibition of topoisomerases [3]. In 2014, approximately 326,000 new substances were discovered by toxicity analyses and pharmacological studies [4]. In particular, plants are a rich resource of secondary metabolites with biological activities [5], for example, the plant-derived compounds bisindole (vinca) alkaloids, camptothecins, epipodophyllotoxins, and taxanes are widely used to treat cancers [1].

*Schisandra chinensis* (*S. chinensis*) is native to Eastern Russia, Northern China, Japan, and Korea [6], and the fruits of this plant have attracted considerable research attention in the medicinal chemistry and drug discovery fields because of their pharmacological activities, which include anti-cancer, anti-viral, anti-inflammatory, antioxidative, and hepatoprotective effects [7,8]. The effects of *S. chinensis* are associated with the dibenzo cyclooctadiene lignans, which include schisandrin, schisantherins A, B, and C, γ-schisandrin, schisanthenol, deoxyschisandrin, and gomisins A and G [9,10]. The structures of these compounds were elucidated using various NMR techniques such as polarization transfer (DEPT), heteronuclear multiple bond correlations (HMBCs), and heteronuclear single quantum coherence (HSQC) [11].

The anti-cancer activities of schisantherins A, B, and C, gomisins A, B, and N, deoxyschisandrin, and α-iso-cubebenol derived from *S. chinensis* have been shown to induce apoptosis, suppress cell cycle arrest, and inhibit the proliferations of various cancer cells, including those of breast, colon, and ovarian cancer and hepatic carcinoma [12,13,14,15,16,17,18,19,20]. However, other compounds present in *S. chinensis* including α-cubebenoate, cubebene, gomisin C, D, E, F, G, K3, and J have not been evaluated for anti-cancer activity in mammalian cells. Among these, α-cubebenoate has a higher probability of having anti-cancer effects because of its structural similarity to α-iso-cubebenol, which has been reported to have therapeutic effects on inflammatory diseases, allergy, sepsis, and obesity [20]. Furthermore, α-cubebenoate was found to remarkably inhibit the expressions of iNOS (inducible nitric oxide synthase) and COX-2 (cyclooxygenase-2), and the productions of NO (nitric oxide) and prostaglandin E2 (PGE2) in mouse peritoneal macrophages [11]. In RBL-2H3 mast cells, α-cubebenoate inhibited antigen-induced degranulation and increased intracellular Ca^2+^ concentrations, and in ovalbumin-challenged BALB/c mice, it suppressed alterations in bronchiolar structure, immune cell accumulations, and the secretions of cytokines by Th2 (T helper 2) cells [21]. Furthermore, in an experimental cecal ligation and puncture (CLP) model α-cubebenoate exhibited anti-septic activity by preventing lung inflammation, increasing phagocytic activity, and reducing inflammatory cytokine levels [22]. Furthermore, in MDI (3-isobutyl-1-methylxanthine, dexamethasone, and insulin)-stimulated 3T3-L1 adipocytes, α-cubebenoate inhibited lipogenesis, stimulated lipolysis, and suppressed inflammasome levels [23]. However, to date, no study has been conducted on the effects of α-cubebenoate on cancers.

The current study was undertaken to evaluate the effects of α-cubebenoate on the viabilities and cell migration and adhesion abilities of CT26 cells (a murine colorectal carcinoma cell line) and CCD-18Co cells (a normal human fibroblast cell line), and to investigate the mechanism responsible.

## 2. Results

### 2.1. Effects of α-Cubebenoate on the Viabilities of CCD-18Co and CT26 Cells

Initially, we investigated whether α-cubebenoate elicited toxic effects in CCD-18Co and CT26 cells using MTT assays after exposing cells to three different concentrations, that is, 7.5 μg/mL α-cubebenoate (Low dose α-cubebenoate, LoCb), 15 μg/mL α-cubebenoate (medium dose α-cubebenoate, MiCb), or 30 μg/mL α-cubebenoate (high dose α-cubebenoate, HiCb) for 24 h. The viability of CT26 cancer cells was remarkably and dose-dependently reduced by α-cubebenoate, and this reduction was slightly higher in the HiCb group than the Cis group (5 μg/mL of cisplatin) (Figure 1b). Furthermore, cell viability tended to decrease with treatment time (Figure 1d). Additionally, these results were in-line with observed changes in cell morphology (Figure 1b). On the other hand, the viability and morphology of CCD-18Co cells were unaffected by α-cubebenoate treatment (Figure 1a). Notably, α-iso-cubebenol and cubebene from *S. chinensis* were not chosen as potential therapeutic candidates because they were toxic to normal and cancer cells (Appendix A). In addition, α-cubebenoate was found to be cytotoxic in HCT116 cells (a human colon cancer cell line) (Figure 1c). Unfortunately, we could not use cancer and normal cells from the same origin since normal murine cells are not commercially available.

### 2.2. Effects of α-Cubebenoate on Apoptosis-Associated Processes in CT26 Cells

To determine whether the cytotoxic effects of α-cubebenoate on CT26 cells involved alterations in apoptosis signaling, we examined the effect of α-cubebenoate on apoptotic cell numbers, MAPK signaling pathway activation, and the expressions of apoptotic proteins. Numbers of apoptotic cells were remarkably increased and numbers of live cells were dose-dependently reduced by α-cubebenoate treatment (Figure 2a). Furthermore, Bax/Bcl-2 ratio and Cleaved Cas-3 expression were dose-dependently increased by α-cubebenoate (Figure 2b). Furthermore, levels of phosphorylated ERK, JNK, and p38 were higher in HiCb treated cells than in Vehicle treated group (Figure 2c).

### 2.3. Effects of α-Cubebenoate on the Migration Ability and Its Associated Signals in CT26 Cells

To determine whether α-cubebenoate suppresses the migration and associated signaling pathways in CT26 cells, changes in wound healing activity and PI3K/AKT-mediated migration activities were investigated in α-cubebenoate treated CT26 cells. The wound-healing assay showed migration was significantly suppressed at an α-cubebenoate concentration of 30 μg/mL (HiCb) (Figure 3a), and a similar pattern was observed in the expression levels of key proteins associated with PI3K/AKT-mediated cell migration (Figure 3b).

### 2.4. Effects of α-Cubebenoate on the Adhesive Ability of CT26 Cells

Finally, we investigated whether α-cubebenoate suppresses the adhesive ability of CT26 cells by assessing the levels of several adhesion-related proteins, that is, VEGFA, MMP2, and MMP9. In addition, the levels of key proteins in the FAK/MLC signaling pathway were assessed in CT26 cells after α-cubebenoate treatment. Levels of VEGFA and MMP9 were remarkably reduced by α-cubebenoate, but MMP2 levels were unaffected (Figure 4a). Additionally, α-cubebenoate suppressed the phosphorylation of key proteins in the FAK/MLC signaling pathway (Figure 4b). However, we did not verify the anti-cancer effects of α-cubebenoate in an experimental animal model with syngeneic tumors because we were unable to acquire sufficient α-cubebenoate.

## 3. Discussion

Several bioactive compounds, such as gomisin N, gomisin A, gomisin J, anwulignan, and α-iso-cubebenol have been identified in *S. chinensis* [16,20,24,25,26], and these observations attracted considerable research attention because of the therapeutic potentials of these compounds. In the present study, we investigated the molecular mechanisms responsible for the anti-cancer effects of α-cubebenoate in CT26 cells. We found α-cubebenoate has notable anti-cancer effects, which included potent cytotoxicity, activation of apoptosis, inhibition of migration, and the suppression of CT26 cell adhesion. Furthermore, our results suggest that α-cubebenoate has potential therapeutic effects against colon cancer.

During investigations on anti-cancer effects, cytotoxicity is perhaps the most important consideration in terms of predicting drug efficacy [27]. In most cases, the deaths of various types of cells are due to apoptosis or necrosis [28]. Of these two types of cell death, apoptosis is considered a target for potential anti-cancer drugs because it induces the death of damaged, non-functional, or outdated cells [29]. Notably, apoptosis is accompanied by the alternation of Bcl-2/Bax expression and MAPK signaling activation in several cancer cells [30]. Some compounds present in *S. chinensis* have also been reported to induce apoptosis. Gomisin N remarkably increased numbers of apoptotic cells and Bax and Bcl-2 protein levels in HepG2 cells [16], and induced apoptosis responses, which included increases in the levels of Cleaved Cas-3 and PARP-1 in HeLa cells. On the other hand, α-iso-cubebenol increased apoptotic cell numbers and Cleaved Cas-3 and Bax levels in HepG2 cells [18,20], and gomisin J enhanced Bax and Cleaved PARP levels in MCF7 and MDA-MB-231 cells [27]. A similar effect was detected in CT26 cells treated with α-cubebenoate. In particular, α-cubebenoate-induced apoptosis was mediated by the alternation of Bax/Bcl-2 pathway and MAPK pathway. Meanwhile, apoptosis and necrosis of cancer cells were caused by the various concentrations of anti-cancer drugs [31]. Since apoptosis has been considered a major mechanism for the cytotoxicity of most anti-cancer drugs, it has received great attention during many preclinical drug discovery investigations [32,33]. Therefore, our study has been focused on analyzing the apoptotic factors to verify the anticancer effects of a-cubebenoate although this analysis can be a limitation of the research. Especially, activation on MAPK signaling pathway in HiCb treated group did not match the alterations in the apoptosis-associated process as shown Figure 2b,c. The level of Cleaved Cas-3 protein was 5-fold higher in Cis group than the HiCb group, while phosphorylation level of ERK, JNK and p38 proteins were 2- to 7-fold higher in the HiCb group than in the Cis group. These differences are thought to be related to MAPK signaling pathway changes because they are known to play important roles in apoptosis, proliferation, stress responses, and differentiation [34].

Previous structure-activity-relationship (SAR) analyses have shown the number and positions of OH groups govern the cytotoxicities of phenolics and that methylation or loss of OH groups is associated with loss of cytotoxicity [35,36]. Furthermore, the cytotoxicity of topoisomerase II inhibitors was provided by OCH_3_ group-based model for cytotoxicity mode of action (MOA) [37]. These relations between molecular features and the cytotoxicity have been reported for some compounds present in *S. chinensis*. Gomisin N with four OCH_3_ groups was reported to be cytotoxic (55%) to HepG2 cells at 320 μM [16], while α-iso-cubebenol with a single OH group was cytotoxic (90%) to HepG2 cells at the same concentration [20]. However, other compounds including gomisin J and anwulignan were cytotoxic at lower concentrations. For example, gomisin J with four OCH_3_ groups and single OH group at 30 μg/mL was cytotoxic to MCF7, MDA-MB-231, MCF10A cells with viabilities 29%, 35%, and 23%, respectively [25], and anwulignan with a single CH_3_O and OH group was cytotoxic to HeLa cells at 60 μM [21]. In the current study, α-cubebenoate at 30 μg/mL (HiCb) with a single OCH_3_ group resulted in a CT26 cells cytotoxicity of ~25% and was more cytotoxic than the other compounds studied,, although anwulignan was more toxic to MDA-MB-231 cells. However, about 25% cytotoxicity to α-cubebenoate can be considered a weak point as one of the anti-tumor drug candidates. To overcome this, it is considered necessary to experiment with high concentrations and long-term treatment of α-cubebenoate.

Metastasis is an extraordinarily complex process whereby cancer cells spread from a primary site to other parts of the body through the blood or lymphatic systems [38]. The metastatic process consists of various steps, which include cancer cell release from primary tumors, migration through surrounding tissues and basement membranes, entry into the circulatory or lymphatic systems or peritoneal space, and settlement and growth in target tissues [39]. In addition, metastasis also mediates various molecular events such as loss of cell-cell adhesion, changes in cell-matrix interactions, degradation of extracellular matrix and basement membranes, and angiogenesis [40,41]. At the molecular level, the PI3K/AKT mediated signaling pathway and the expressions of related proteins such as MMPs and VEGF are important regulators of migration and invasion [42,43]. Several plant-derived compounds inhibit cancer cell metastasis by regulating molecular events. For example, resveratrol, which is found in peanuts, soybeans, purple grapes, and pomegranates, inhibits pancreatic cancer cells by regulating *N*-cadherin [44], and epigallocatechin-3-gallate (EGCG) found in green tea (*Camellia sinensis*, Theaceae) and genistein first isolated from *Genista tinctoria* suppressed metastatic activity by downregulating MMP-2 expression in some types of cancers in melanoma, breast cancer and prostate cancer [45,46,47,48]. Furthermore, boswellic acid from *Boswellia serrata* and bromelain from *Ananas comosus* showed anti-angiogenic activity by interfering with the VEGF signaling pathway [49,50,51]. However, no compound identified in *S. chinensis* had been investigated with respect to its effects on the metastasis of mammalian cancer cells. Therefore, our findings provide the first evidence that α-cubebenoate (a component of *S. chinensis*) suppresses cancer cell metastasis. We believe that the inhibitory effects of α-cubebenoate on CT26 cancer growth are associated with the inhibition of adhesion ability via regulation of the FAK/MLC signaling pathway and with the suppression of migration ability via regulation of the PI3K/AKT signaling pathway.

## 4. Materials and Methods

### 4.1. Preparation of α-Cubebenoate

α-Cubebenoate was isolated from the dried fruits of *S. chinensis* by sequential extraction using ethyl alcohol (EtOH), chloroform (CHCl_3_), methyl alcohol (MeOH), and hexane [11]. About 20.8 mg of α-cubebenoate was extracted per kilogram of dried fruit [6].

### 4.2. Cell Culture

CT26 cancer cells (a murine colorectal carcinoma cell line derived from the colon tissues of a BALB/c mouse) were purchased from the ATCC (Cat. No. CRL-2638, Manassas, VA, USA). This cell line was selected because it has been widely used in previous studies on the anti-cancer effects of natural products. HCT116 cells (a colorectal carcinoma cell line derived from the human colon tissues) were purchased from the ATCC (Cat. No. CCL-247). CCD-18Co cells were derived from normal human colon tissue and were also purchased from the ATCC (Cat No. CRL-1459). CT26 cells and HCT116 cells were cultured in Roswell Park Memorial Institute 1640 Medium (RPMI 1640 Medium, Cat. No. LM011-01, Welgene, Gyeongsan-si, Korea) supplemented with 10% fetal bovine serum (FBS, Welgene, Gyeongsan-si, Korea), 2 mM of glutamine, 100 U/mL of penicillin, and 100 μg/mL of streptomycin (Thermo Fisher Scientific Inc., Wilmington, DE, USA), and CCD-18Co cells were cultured in Dulbecco Modified Eagle’s Medium (DMEM, Cat. No. LM001-05, Welgene) supplemented with 10% fetal bovine serum (FBS, Welgene, Gyeongsan-si, Korea), L-glutamine, penicillin, and streptomycin (Thermo Fisher Scientific Inc., Wilmington, DE, USA). Both cell lines were cultured in a humidified 5% CO_2_ and 95% air incubator at 37 °C.

### 4.3. Cytotoxicity Assay

The viabilities of CT26, HCT116, and CCD-18Co cells were assessed using an MTT (3-[4,5-dimethylthiazol-2-yl]-2,5-diphenyltetrazolium bromide) assay (Cat. No. M2128, Sigma-Aldrich Co., St. Louis, MO, USA). The anti-cancer activity of α-cubebenoate isolated from *S. chinensis* fruit was determined as previously described for gomisin J [25]. Additionally, α-cubebenoate used in this assay was decided based on the results of previous studies on the anti-obesity activity of α-cubebenoate and the anti-cancer activity of gomisin A [13,23]. Briefly, the two cell types were evenly seeded at a density of 7 × 10^3^ cells/200 μL in RPMI 1640 and grown in a 5% CO_2_ incubator for 24 h at 37°C. On attaining 70–80% confluence, cells were either treated with Vehicle (dimethyl sulfoxide (DMSO, Cat. No. D1370.0100, Duchefa Biochemie, Haarlem, The Netherlands), Vehicle treated group), pretreated with 5 μg/mL of cisplatin (Cis, positive control group), 7.5 μg/mL α-cubebenoate (Low dose α-cubebenoate, LoCb), 15 μg/mL α-cubebenoate (Medium dose α-cubebenoate, MiCb), or 30 μg/mL α-cubebenoate (High dose α-cubebenoate, HiCb) dissolved in DMSO. The α-cubebenoate used was donated by Professor Young Whan Choi (one of the authors) [39]. Cis dosages were determined based on a preliminary experiment involving IC_50_ of Cis = 4.28 μg/mL (Appendix A) and a previous study that suggested an effective concentration of Cis (from 4.5 to 45 μg/mL) [52]. α-Cubebenoate doses were determined by preliminary experimentation (Appendix A) and a previous study in which concentrations from 10 to 30 μg/mL were used [23]. After incubating CT26 cells for 24 h, 200 μL of fresh RPMI 1640, and 50 μL of MTT solution (2 mg/mL in 1× PBS) were added to each well. Following incubation for 4 h at 37°C, formazan precipitates were completely dissolved in DMSO, and absorbance was read at 570 nm using a Vmax plate reader (Molecular Devices, Sunnyvale, CA, USA). Cell morphologies were observed under an optical microscope (Leica Microsystems, Heerbrugg, Switzerland) at 200× magnification.

### 4.4. Analysis of Apoptotic Cells by Fluorescence-Activated Cell Sorting (FACS)

Apoptotic cell percentages were determined using the Muse™ Annexin V and Dead Cell Kit (Cat. No. MCH100105, Millipore Co., Billerica, MA, USA). Briefly, CT26 cells were harvested after incubation with 7.5 (LoCb), 15 (MiCb), or 30 (HiCb) μg/mL of α-cubebenoate or 5 μg/mL of cisplatin for 24 h, suspended in RPMI 1640 (3.5 × 10^4^ cells/mL), and then a cell suspension (100 μL, 1 × 10^5^ cells/mL) was incubated with annexin V and 7-aminoactinomycin D (7-AAD) (kit components) for 20 min at room temperature. Finally, cells were analyzed using a Muse™ Cell Analyzer (Cat. No. PB4455ENEU, Millipore Co.). After gating for cell size, cells were classified as; non-apoptotic (Annexin V (−) and 7-AAD (−)), early apoptotic (Annexin V (+) and 7-AAD (−)), late apoptotic (Annexin V (+) and 7-AAD (+)), or as nuclear debris (Annexin V (+) and 7-AAD (+)).

### 4.5. Wound-Healing Assay

The wound-healing assay used was as previously described [53]. Briefly, CT26 cells were evenly seeded in 6-well plates and grown to 80–90% confluence. After removing the culture medium, a wound was created using a sterile pipette tip in each plate, and detached cells and cellular debris were removed by washing twice with 1× PBS buffer. The attached cells were incubated for 9 or 18 h in RPMI 1640 containing 7.5 (LoCb), 15 (MiCb), or 30 (HiCb) μg/mL of α-cubebenoate or 5 μg/mL of Cis. Cell migration into wounds was photographed at two different time points (9 and 18 h after wounding) using a microscope (Leica Microsystems) at 100× magnification. Wound closure rates were calculated using the following formula:Wound closure rate (%) = (Original width−Width after migration)/Original width × 100

### 4.6. Western Blot

Total homogenates of CT26 cells were collected using the Pro-Prep Protein Extraction Solution (Cat. No. 17081, Intron Biotechnology Inc., Seongnam, Korea) and centrifuged at 15,000 rpm for 10 min. Total protein concentrations were determined using the SMART™ Bicinchoninic Acid Protein Assay Kit (Cat. No. 23225, Thermo Fisher Scientific Inc.), and 30 μg quantities were subjected to 4–20% SDS-PAGE for 2 h. Proteins were transferred to 0.45 μm nitrocellulose blotting membranes (Cat. No. 10600003, GE Healthcare, Little Chalfont, UK) for 2 h at 40 V, and membranes were then incubated separately with specific primary antibodies {anti-Bax [E63] (Cat. No. ab32503, Abcam, Cambridge, UK), anti-Bcl-2 (Cat. No. PA5-20069, Thermo Fisher Scientific Inc.), anti-Cas-3 (Cat. No. 9662S, Cell Signaling Technology, Danvers, MA, USA), anti-p44/42 MAPK (ERK1/2) (Cat. No. 9102S, Cell Signaling Technology), anti-p-ERK (E-4) (Cat. No. 9101S, Cell Signaling Technology), anti-JNK (Cat. No. 9252S, Cell Signaling Technology), anti-p-JNK (Cat. No. 9251S, Cell Signaling Technology), anti-p38 (Cat. No. 9212S, Cell Signaling Technology), anti-p-p38 (Cat. No. 9211S, Cell Signaling Technology), anti-PI3K (Cat. No. 4292S, Cell Signaling Technology), anti-p-PI3K (Cat. No. 4228S, Cell Signaling Technology), anti-AKT (Cat. No. 9272S, Cell Signaling Technology), anti-p-AKT (Cat. No. 4058S, Cell Signaling Technology), anti-VEGFA (Cat. No. ab46154, Abcam), anti-MMP-2 (H-76) (Cat. No. SC-10736, Santa Cruz Biotechnology), anti-MMP-9 (G657) (Cat. No. 2270S, Cell Signaling Technology), anti-FAK (Cat. No. 3285S, Cell Signaling Technology), anti-p-FAK (Cat. No. 3283S, Cell Signaling Technology), anti-MLC (Cat. No. ab92721, Abcam), anti-p-MLC (Cat. No. ab2480, Abcam), or anti-β-actin antibodies (Cat. No. 4967S, Cell Signaling Technology)} overnight at 4°C. Probed membranes were then washed with standard washing buffer, incubated with 1:1000 diluted horseradish peroxidase (HRP)-conjugated goat anti-rabbit IgG (Cat. No. G21234, Thermo Fisher Scientific Inc.) for 1 h, and developed using Amersham™ ECL Select™ Western Blotting detection reagent (Cat. No. RPN2235, GE Healthcare). Finally, chemiluminescence signals of specific protein bands were quantified using a FluorChemi^®^FC2 densitometer (Alpha Innotech Co., San Leandro, CA, USA).

### 4.7. Statistical Analysis

The significances of intergroup differences were determined by One-way Analysis of Variance (ANOVA) (SPSS for Windows, Release 10.10, Standard Version, Chicago, IL, USA) followed by the Tukey post hoc t-test for multiple comparisons. Results are presented as means ± SDs, and statistical significance was accepted for *p* values < 0.05.

## 5. Conclusions

The results of the present study provide novel scientific evidence that α-cubebenoate has anti-cancer activity, and that this activity is associated with its cytotoxic and apoptotic effects, suppression of migration ability, and inhibition of cell adhesion. Furthermore, the study shows α-cubebenoate has potential use as an anti-cancer drug without accompanying side effects. However, further studies are needed to advance our understanding of the anti-cancer effects of α-cubebenoate and the molecular mechanisms responsible for these actions.

## Figures and Tables

**Figure 1 molecules-27-00737-f001:**
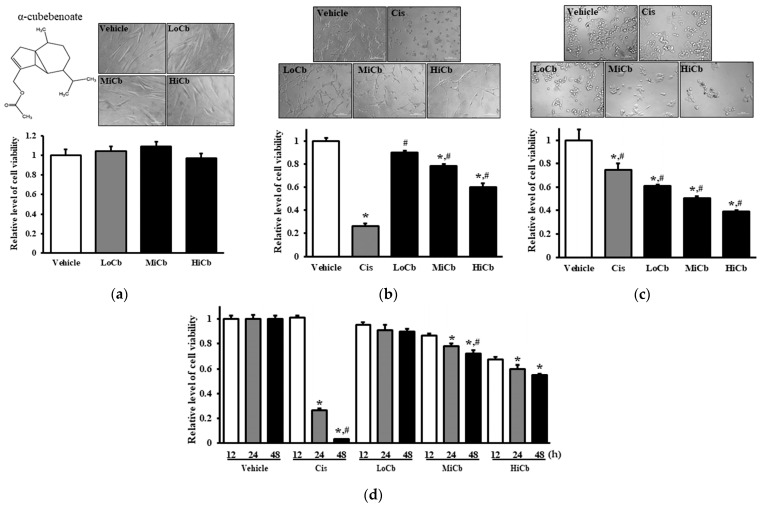
Cytotoxic effects of α-cubebenoate treated CCD-18Co (**a**), CT26 cells (**b**), and HCT116 cells (**c**). After incubation of CCD-18Co, CT26, or HCT116 cells with 7.5 (LoCb), 15 (MiCb) and 30 (HiCb) μg/mL of α-cubebenoate or Cis for 24 h, morphological changes were observed under a microscope at 200× magnification. (**d**) Furthermore, time-dependent response to 7.5 (LoCb), 15 (MiCb) and 30 (HiCb) μg/mL of α-cubebenoate was assessed in CT26 cells after treatment for 12, 24, or 48 h. Two to three wells per group were used for the MTT assay, and optical densities were measured in duplicate. Data are presented as means ± SDs. *, *p* < 0.05 versus Vehicle treated group. #, *p* < 0.05 versus Cis treated group.

**Figure 2 molecules-27-00737-f002:**
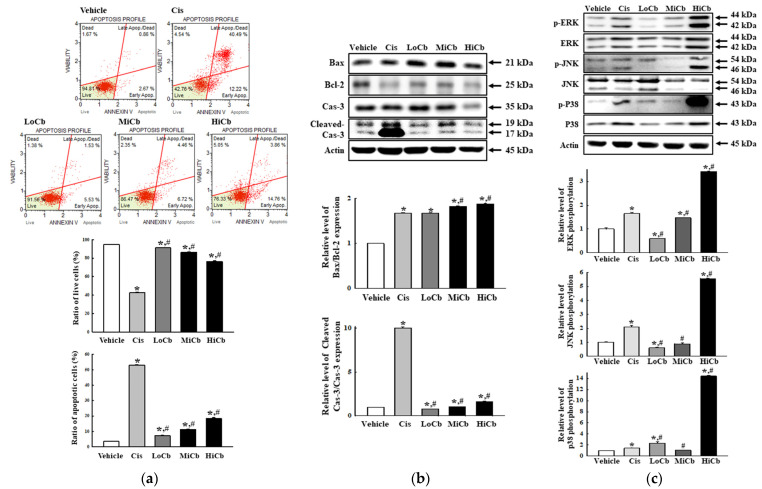
Apoptosis of α-cubebenoate treated CT26 cells. (**a**) Analysis of annexin V and 7-AAD stained CT26 cells. After treatment with 7.5 (LoCb), 15 (MiCb), or 30 (HiCb) μg/mL of α-cubebenoate for 24 h, cell distributions were analyzed by annexin V and 7-AAD staining. Gating of initial cell population is placed on cell size vs. annexin V. Sequentially, the most obvious debris was excluded from total cell population. Two to three wells per group were used for annexin V and 7-AAD staining, and numbers of dead and live cells were measured in duplicate. (**b**) Expressions of apoptotic proteins. After treatment with 7.5 (LoCb), 15 (MiCb), or 30 (HiCb) μg/mL of α-cubebenoate for 24 h, the expression levels of Bax, Bcl-2, Cas-3 and Cleaved Cas-3 proteins were determined using an imaging densitometer. (**c**) Expressions of MAPK signaling pathway members. After treatment with 7.5 (LoCb), 15 (MiCb), or 30 (HiCb) μg/mL of α-cubebenoate for 24 h, the expression levels of ERK, p-ERK, JNK, p-JNK, p38 and p-p38 proteins were determined using an imaging densitometer. Two or three dishes per group were to prepare cell homogenates, and Western blot analysis was performed in duplicate. Data are reported as means ± SDs. *, *p* < 0.05 versus Vehicle treated group. #, *p* < 0.05 versus the Cis treated group.

**Figure 3 molecules-27-00737-f003:**
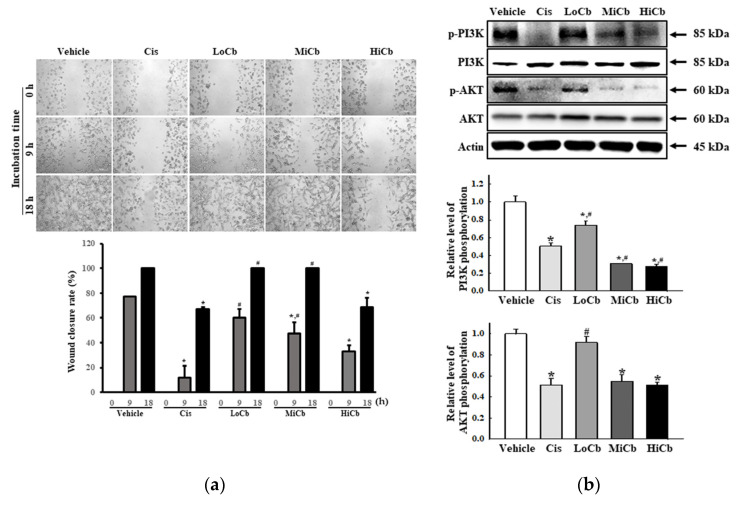
Cell migration analysis of α-cubebenoate treated CT26 cells. (**a**) Wound healing assay. The migration ability of CT26 cells was analyzed using the wound healing assay after treatment with 7.5 (LoCb), 15 (MiCb), or 30 (HiCb) μg/mL of α-cubebenoate. Images of cells were captured after incubation for 9 h or 18 h at 100× magnification. Two to three wells per group were used for the wound healing assay, and closure rates were calculated in duplicate. (**b**) Expressions of PI3K/AKT signaling pathway members. After treatment with 7.5 (LoCb), 15 (MiCb), or 30 (HiCb) μg/mL of α-cubebenoate for 24 h, the expression levels of PI3K, p-PI3K, AKT and p-AKT proteins were determined using an imaging densitometer. Two to three dishes per group were used to prepare cell homogenates, and Western blot analysis was performed in duplicate. Data are reported as means ± SDs. *, *p* < 0.05 versus Vehicle treated group. #, *p* < 0.05 versus the Cis treated group.

**Figure 4 molecules-27-00737-f004:**
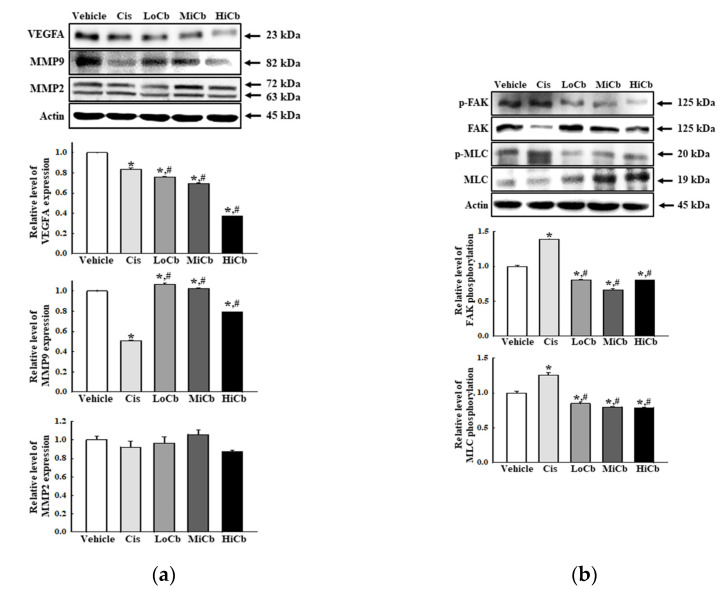
Cell adhesion ability of α-cubebenoate treated CT26 cells. (**a**) Expression of cell adhesion proteins. After treatment with 7.5 (LoCb), 15 (MiCb), or 30 (HiCb) μg/mL of α-cubebenoate for 24 h, the expression levels of VEGFA, MMP2, and MMP9 proteins were determined using an imaging densitometer. (**b**) Expression of FAK/MLC signaling pathway members. After treatment with 7.5 (LoCb), 15 (MiCb), or 30 (HiCb) μg/mL of α-cubebenoate for 24 h, the expression levels of FAK, p-FAK, MLC, and p-MLC proteins were determined using an imaging densitometer. Two to three dishes per group were used to prepare cell homogenates, and Western blot analysis was performed in duplicate. Data are reported as the means ± SDs. *, *p* < 0.05 versus Vehicle treated group. #, *p* < 0.05 versus the Cis treated group.

## Data Availability

All the data that support the findings of this study are available on request from the corresponding author.

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
