# Peer review of "Anti-Cancer Effects of α-Cubebenoate Derived from *Schisandra chinensis* in CT26 Colon Cancer Cells"

_molecules, 2022, doi:10.3390/molecules27030737_

Round 1

Reviewer 1 Report

The objective of this study was to assess in vitro anti-tumor activity of α-cubebenoate derived from Schisandra chinensis in CT26 colon cancer cells. In my opinion the study is valuable and the presented results are convincing. The paper is concise and well written.

There are only few minor points which should be addressed by the authors:

- why did the authors choose these particular cell lines?

- why just two to three wells per group were used in the MTT assay?

- Discussion, lines 177-178: “our results suggest that α-cubebenoate should be considered a novel drug candidate for colon cancer” – this sentence must be weighed up.

Author Response

The objective of this study was to assess in vitro anti-tumor activity of α-cubebenoate derived from Schisandra chinensis in CT26 colon cancer cells. In my opinion the study is valuable and the presented results are convincing. The paper is concise and well written.

- Why did the authors choose these particular cell lines?

☞ The reasons for the selection were added into materials and methods section (line 316-319) as followings;

“This cell line was selected because it is widely used in previous studies on the anti-cancer effects of various natural products.”

- Why just two to three wells per group were used in the MTT assay?

☞ In MTT assay, cell viability analyses were repeated using two to three wells per group.  This analysis was sufficient to obtain statistical significance. Please understand this situation.

- Discussion, lines 177-178: “our results suggest that α-cubebenoate should be considered a novel drug candidate for colon cancer” – this sentence must be weighed up.

☞ That has been corrected in line 219-221 as following;

“Furthermore, our results suggest that α-cubebenoate has potential therapeutic effects against colon cancer.”

Reviewer 2 Report

The manuscript submitted by Gong et,al. investigated the potential anti-cancer effect of α-cubebenoate derived from Schisandra chinensis in CT26 colon cancer cells. The investigators tried to assess cytotoxic, apoptotic and anti-migration effect of α-cubebenoate on CT26 cell line.

This study has major issues:

  1. The study was designed using one cancer cell line (CT26 cell line) which is a murine based cell line. This study should have included more than one cell line (especially human based cell lines e.g. HCT116, HT29…etc), so any conclusion based on one cell line is not satisfactory.
  2. The cytotoxic effect of α-cubebenoate should have been evaluated in dose dependent and time dependent manner. In this study, the investigator chose to use three concentration of α-cubebenoate based on a previous study on 3T3L1 cells which showed that these concentrations were not toxic on 3T3L1 cells, I can not understand the justification behind this. The investigators should have done cytotoxicity evaluation using serial dilutions and for different durations, so they can calculate the IC50 of α-cubebenoate and figure out if the effect is also time dependent, and based on these results they can determine the concentrations to use.
  3. Furthermore, and even at the highest concentration used in this study (30µg/ml) the cytotoxic effect did not exceed 25%, and this considered a relatively weak cytotoxic effect.
  4. On what basis did the investigator choose the concentration of cisplatin used (5 µg/ml), based on the literature the IC50 of Cisplatin on CT26 cell line is not consistent, accordingly, the authors should have conducted a serial dilution to find out the IC50 of cisplatin in order to be used in their experimental set up.
  5. When it comes to the effect of HiCB on ERK and JNK phosphorylation as shown in 2.C., there was a significant increase in ERK and JNK phosphorylation even when compared to cisplatin, this is unexpected and does not fit with other effects (when compared to cisplatin), in addition the authors did not provide any interpretation or explanation of  these finding at all in their discussion.
  6. When it comes to the effect of α-cubebenoate on PI3K and AKT phosphorylation, HiCB and MiCB inhibited Pi3K phosphorylation even more than cisplatin or similar to cisplatin when it comes to AKT phosphorylation, this is unexpected, and was not discussed by the authors.
  7. The authors should include the source of α-cubebenoate in their material and methods
  8. Discussion required an intensive revision, most of the findings were not discussed and they focused on structural activity relationship (SAR), which is not highly relevant. They should have tried to go over each finding and try to discuss it thoroughly with a most possible supported interpretation.
  9. Introduction section need also an intensive revision, ideas are not connected
  10. The manuscript requires an extensive English editing, there are many grammatical mistakes, the ideas are not connected.
  11. The conclusion of this not study is not supported by the findings, based on the finding α-cubebenoate does not have a remarkable anticancer effect at least in the experimental setup used.

Author Response

  1. The study was designed using one cancer cell line (CT26 cell line) which is a murine based cell line. This study should have included more than one cell line (especially human based cell lines e.g. HCT116, HT29…etc), so any conclusion based on one cell line is not satisfactory.

☞ According to your comments, we have added new results using a different cell line in line 139-141 as following;

“In addition, α-cubebenoate was found to be cytotoxic in HCT116 cells (a human colon cancer cell line) (Figure 1d).”

  1. The cytotoxic effect of α-cubebenoate should have been evaluated in dose dependent and time dependent manner. In this study, the investigator chose to use three concentration of α-cubebenoate based on a previous study on 3T3L1 cells which showed that these concentrations were not toxic on 3T3L1 cells, I can not understand the justification behind this. The investigators should have done cytotoxicity evaluation using serial dilutions and for different durations, so they can calculate the IC50 of α-cubebenoate and figure out if the effect is also time dependent, and based on these results they can determine the concentrations to use.

☞ According to your comments, Figure 1(c) has added the new results for cytotoxicity of α-cubebenoate evaluated at three different time points (12, 24 and 48 h) in line 118-119 as following;

“Furthermore, cell viability tended to decrease with treatment time (Figure 1c).”

Also, the reasons for the dosage selection were added into materials and methods section (line 347-349) as followings;

“α-Cubebenoate doses were determined by preliminary experimentation (Supplementary Figure S3) and a previous study in which concentrations from 10 to 30 μg/mL were used [23].”

  1. Furthermore, and even at the highest concentration used in this study (30µg/ml) the cytotoxic effect did not exceed 25%, and this considered a relatively weak cytotoxic effect.

☞ This is good point. This issue has been further described in Discussion part (line 274-277) as following;

“However, about 25% cytotoxicity to α-cubebenoate can be considered a weak point as one of anti-tumor drug candidates. To overcome this, it is considered necessary to experiment with high concentrations and long-term treatment of α-cubebenoate.”

  1. On what basis did the investigator choose the concentration of cisplatin used (5 µg/ml), based on the literature the IC50 of Cisplatin on CT26 cell line is not consistent, accordingly, the authors should have conducted a serial dilution to find out the IC50 of cisplatin in order to be used in their experimental set up.

☞ According to your comments, we have determined the IC50 of Cisplatin and inserted as Supplement Figure S2 in Materials and methods (line 346-348) as following;

“Cis dosages were determined based on preliminary experiment involving IC50 of Cis = 4.28 μg/mL (Supplementary Figure S2) and a previous study that suggested an effective con-centration of Cis (from 4.5 to 45 μg/mL) [52].”

  1. When it comes to the effect of HiCB on ERK and JNK phosphorylation as shown in 2.C., there was a significant increase in ERK and JNK phosphorylation even when compared to cisplatin, this is unexpected and does not fit with other effects (when compared to cisplatin), in addition the authors did not provide any interpretation or explanation of these finding at all in their discussion.

☞ According to your comments, this issue has been further described in Discussion part (line 246-253) as following;

“Especially, activation on MAPK signaling pathway in HiCb treated group did not match the alterations in apoptosis-associated process as shown Figure 2b and c. The level of cleaved Cas-3 protein was 5-fold higher in Cis group than the HiCb group, while phosphorylation level of ERK, JNK and p38 proteins were 2- to 7-fold higher in the HiCb group than in the Cis group. These differences are thought to be related to MAPK signaling pathway changes, because they are known to play important roles in apoptosis, proliferation, stress responses, and differentiation [34].”

  1. When it comes to the effect of α-cubebenoate on PI3K and AKT phosphorylation, HiCB and MiCB inhibited Pi3K phosphorylation even more than cisplatin or similar to cisplatin when it comes to AKT phosphorylation, this is unexpected, and was not discussed by the authors.

☞ This is good point. However, there was no significant statistical difference in the level of PI3K and AKT phosphorylation between the Cis treated group and the HiCb/MiCb treated group. Please understand this situation.

  1. The authors should include the source of α-cubebenoate in their material and methods

☞ According to your comments, we have added the section for the source of α-cubebenoate (line 309-313) as following;

 “4.1. Preparation of α-cubebenoate

α-Cubebenoate was isolated from the dried fruits of S. chinensis by sequential extraction using ethyl alcohol (EtOH), chloroform (CHCl3), methyl alcohol (MeOH), and hexane [11]. Finally, about 20.8 mg of α-cubebenoate was extracted per kilogram of dried fruit  from the dried fruits of S. chinensis (1.0 kg)[6]. ”

  1. Discussion required an intensive revision, most of the findings were not discussed and they focused on structural activity relationship (SAR), which is not highly relevant. They should have tried to go over each finding and try to discuss it thoroughly with a most possible supported interpretation.

☞ According to your comments, we has added some paragraph in Discussion part (line 246-253 and 291-299) as following;

“Especially, activation on MAPK signaling pathway in HiCb treated group did not match the alterations in apoptosis-associated process as shown Figure 2b and c. The level of cleaved Cas-3 protein was 5-fold higher in Cis group than the HiCb group, while phosphorylation level of ERK, JNK and p38 proteins were 2- to 7-fold higher in the HiCb group than in the Cis group. These differences are thought to be related to MAPK signaling pathway changes, because they are known to play important roles in apoptosis, proliferation, stress responses, and differentiation [34].”

“Several plant-derived compounds inhibit cancer cell metastasis by regulating molecular events. For example, resveratrol, which is found in peanuts, soybeans, purple grapes, and pomegranates, inhibits pancreatic cancer cells by regulating N-cadherin [44], and epi-gal-locatechin-3-gallate (EGCG) found in green tea (Camellia sinensis, Theaceae) and genistein first isolated from Genista tinctoria suppressed metastatic activity by downregulating MMP-2 expression in some types of cancers in melanoma, breast cancer and prostate cancer [45-48]. Furthermore, boswellic acid from Boswellia serrata and bromelain from Ananas comosus showed anti-angiogenic activity by interfering with the VEGF signaling pathway [49-51].”

  1. Introduction section need also an intensive revision, ideas are not connected

☞ According to your comments, we has added some paragraph into Introduction part (line 71-79) as following;

“However, other compounds present in S. chinensis including α-cubebenoate, cubebene, gomisin C, D, E, F, G, K3, and J have not been evaluated for anti-cancer activity in mam-malian cells. Among these, α-cubebenoate has a higher probability of having for anti-cancer effects because of its has a structural similarity to cubebenol, which has been shown to have therapeutic effects on inflammatory diseases, allergy, sepsis, and obesity [20]. Furthermore, α-cubebenoate was found to remarkably inhibited the expressions of iNOS (inducible nitric oxide synthase) and COX-2 (cyclooxygenase-2), and the productions of NO (nitric oxide) and prosta-glandin E2 (PGE2) in mouse peritoneal macrophages [11].”

  1. The manuscript requires an extensive English editing, there are many grammatical mistakes, the ideas are not connected.

☞ This manuscript has been edited by professional company and a proof of calibration is attached. 

  1. The conclusion of this not study is not supported by the findings, based on the finding α-cubebenoate does not have a remarkable anticancer effect at least in the experimental setup used.

☞ This is good point. Actually, our study didn't show dramatic anti-cancer effects of α-cubebenoate in CT26 cells. However, these evidences were enough to show a novel function of α-cubebenoate as other compounds derived from plants in most previous studies. Also, the cisplatin were used as positive controls to prove the anti-cancer effects of a-cubebenate. Furthermore, some paragraphs have been corrected or added in our manuscript to solve this issue and improve the quality of scientific evidence. Please understand this situation.

Reviewer 3 Report

The manuscript present an interesting research on α-cubebenoate, a natural compound isolated from Schisandra chinensis fruits. The compounds identified in this medicinal plant have been shown to be substances with important pharmacological properties such as anti-inflammatory, hepatoprotective, antiviral, antitumor etc. The scientific value of the presented work is raised by testing a compound isolated from plant material, which in previous studies has shown anti-allergic, anti-obesity and anti-inflammatory properties. Authors obtained interesting results that could be useful for further studies, and the tested compound could be a promising candidate for the anticancer therapy.

The research was well conducted, but I did not understand the following affirmation:

Line 215-216: “In the case of α-cubebenoate used in these studies, cytotoxicity is considered to have worked by the OCH3 group rather than the OH group.’’, provided that no OCH3 and OH groups appear in the chemical structure of α-cubebenoate.

Author Response

Line 215-216: “In the case of α-cubebenoate used in these studies, cytotoxicity is considered to have worked by the OCH3 group rather than the OH group.’’, provided that no OCH3 and OH groups appear in the chemical structure of α-cubebenoate.

☞ This is probably because the size of the figure image is small. Therefore, we have enlarged the size of the figure to clearly see the structure of each chemical group.

Reviewer 4 Report

The manuscript entitled “Anti-tumor effects of α-Cubebenoate derived from Schisandra chinensis in CT26 colon cancer cells” aimed to investigate potential anti-tumor activity of α-cubebenoate against colon carcinoma, alterations in the proliferation, apoptosis, and metastasis of CT26 cells. The viabilities of cancer cells were decreased in dose-dependent way, whereas a normal human fibroblast cells stayed unaffected.  Besides, α-cubebenoate increased the number of apoptotic CT26 cells and increased Bax, Bcl-2, Cas-3 and cleaved Cas-3 protein levels by activating the MAP kinase signalling pathway as well as suppressed CT26 migration. Authors concluded that α-cubebenoate could stimulate apoptosis and inhibit metastasis by regulating several signalling pathway.

Authors changed their manuscript accordingly but there are few minor remarks that needs to be resolved.

Page 1, line 37 – please rephrase “These compounds are enhanced the resultant biological diversity because they contain chiral centers, complex ring systems, and certain heteroatoms”

Page 7, line 194 – please change to “Since apoptosis has been considered as major mechanism for the cytotoxicity of most anticancer drugs, it has received a great attention during many preclinical drug discovery investigations”

Page 7, line 197 – please change to “Therefore, our study has been focused on analyzing the apoptotic factors to verify the anticancer effects of a-cubebenoate although this analysis can be a limitation of research.”

Page 7, 219 – please change to “The metastatic process consists of various steps, which includes cancer cell release from primary tumors, migration through surrounding tissues and basement membranes, entry into the circulatory or lymphatic systems or peritoneal space, and settlement and growth in target tissues”

Page 9, line 333 – please change to “Moreover, our study was not conducted using tumor and normal cells of the same origin since the normal murine cells are not commercially available.”

Author Response

Page 1, line 37 – please rephrase “These compounds are enhanced the resultant biological diversity because they contain chiral centers, complex ring systems, and certain heteroatoms”

☞ According to your comments, they have been corrected in line 41-43 as following;

“The chiral centers, complex ring systems, and certain heteroatoms of these compounds causes the enhanced of the resultant biological diversity [2].”

Page 7, line 194 – please change to “Since apoptosis has been considered as major mechanism for the cytotoxicity of most anticancer drugs, it has received a great attention during many preclinical drug discovery investigations”

☞ They have been corrected in line 239-241.

Page 7, line 197 – please change to “Therefore, our study has been focused on analyzing the apoptotic factors to verify the anticancer effects of a-cubebenoate although this analysis can be a limitation of research.”

☞ They have been corrected in line 243-245.

Page 7, 219 – please change to “The metastatic process consists of various steps, which includes cancer cell release from primary tumors, migration through surrounding tissues and basement membranes, entry into the circulatory or lymphatic systems or peritoneal space, and settlement and growth in target tissues”

☞ They have been corrected in line 280-284.

Page 9, line 333 – please change to “Moreover, our study was not conducted using tumor and normal cells of the same origin since the normal murine cells are not commercially available.”

☞ They have been corrected in line 126-128 as following;

“Unfortunately, we could not use cancer and normal cells from the same origin since normal murine cells are not commercially available.”

Reviewer 5 Report

My comments on the article remained the same. Despite the fact that I still believe that this material is of interest and can be published, I have complaints about the placement (distribution) of the material within the article. The discussion section should give an idea of ​​the content of the article, while the authors' main content is shifted to the Results section. In the results section, only specific facts should be given, and their analysis (comparison of trends and correlations) is moved to the Discussion section.
In addition, the last two sentences from the Conclusion section (on the insufficient amount of substance and on the inaccessibility of normal cells of the type required by the author) should be moved to the results section.

Author Response

My comments on the article remained the same. Despite the fact that I still believe that this material is of interest and can be published, I have complaints about the placement (distribution) of the material within the article. The discussion section should give an idea of ​​the content of the article, while the authors' main content is shifted to the Results section. In the results section, only specific facts should be given, and their analysis (comparison of trends and correlations) is moved to the Discussion section.

☞ According to your comments, we have modified the Results section to leave only facts.

In addition, the last two sentences from the Conclusion section (on the insufficient amount of substance and on the inaccessibility of normal cells of the type required by the author) should be moved to the results section.

☞ According to your comments, they were transferred into Results section (line 196-198) and (line 126-128).

Round 2

Reviewer 2 Report

I would like to thank the authors for their efforts, they tried to address my comments and suggestions as much as possible. They included some new data to enhance the story and I think now it is significantly improved.

This manuscript is a resubmission of an earlier submission. The following is a list of the peer review reports and author responses from that submission.

Round 1

Reviewer 1 Report

  1. A large number of literatures have reported antitumor activities, but the authors of this article have not reviewed the previous findings, suggesting that the introduction should be added to other people's antitumor drug research progress.
  2. MiCb seems to be dying more than HiCb IN Fig1(b) , but the statistical analysis shows that HiCb is dying more than HiCb, so it doesn't match the image.
  3. The cleavage-cas3 in Fig2 is not clear in vehicle and HiCb, so it is difficult to analyze the result.
  4. The bands of the phosphorylated Pi3k in Cis group of Fig. 3(b) appear to be significantly shallower than those of LoCb, but the results of statistical analysis were similar. So it is suspected that errors occur in the statistical process.
  5. In Fig. 3(a) the control group did not appear to grow very well for 9 hours, but the statistical result was 100% full, whether there was an error or not.
  6. In Fig4(a) MMP9 CIS bands and (b) P-MLC and MLC bands are not clear, so it is difficult to analyze the results.
  7. Although the in vitro and in Vivo experiments show that the drug has an effect on the proliferation of tumor cells, anti-tumor experiments should be carried out in Vivo to prove the effectiveness of the drug.
  8. P value should be in Italics

Reviewer 2 Report

The manuscript entitled “Anti-tumor effects of α-Cubebenoate derived from Schisandra chinensis in CT26 colon cancer cells” aimed to investigate potential anti-tumor activity of α-cubebenoate against colon carcinoma, alterations in the proliferation, apoptosis, and metastasis of CT26 cells. The viabilities of cancer cells were decreased in dose-dependent way, whereas a normal human fibroblast cells stayed unaffected.  Besides, α-cubebenoate increased the number of apoptotic CT26 cells and increased Bax, Bcl-2, Cas-3 and cleaved Cas-3 protein levels by activating the MAP kinase signalling pathway as well as suppressed CT26 migration. Authors concluded that α-cubebenoate could stimulate apoptosis and inhibit metastasis by regulating several signalling pathway.

In the Introduction section, in the last paragraph, please provide the full type of cancer cells used in the study on the first mention e.g. murine colorectal carcinoma cell line. Please also mention that the studies are done on normal cells e.g. CCD-18Co cells.

Please also provide details of what is actually investigated and what methods are used in the same paragraph e.g. cytotoxicity, type of cell death, adhesive ability etc.

Please explain low, medium and high doses of selected agent, on what is that based? Are they sub-toxic, toxic, etc. Explain also rational for using selected dosed in your experiments.

In the discussion section, authors are stating that apoptosis is considered a target for potential anti-tumor drugs because it induces the death of damaged, non-functional, or outdated cells. Although decades of research clarified the pathways that regulate apoptosis and allowed the development of novel diagnostic and therapeutic modalities in cancer treatment, only recently has the significance of necrosis become the focus of investigations as well. Necrosis is an irreversible inflammatory form of cell death with a possible

implication for cancer therapy. Although necrosis was viewed as strictly a pathologic form of cell death that is not a physiologically programmed process there are large number of experimental data that indicate that, much like apoptosis, specific genes have evolved to regulate necrotic cell death, suggesting that necrosis may be a well-regulated process activated by rather specific physiological and pathological stimuli, hence, necrosis is not to be excluded as a possible way of cancer cell death. Please address this issue.

Reviewer 3 Report

The originality of this paper as well as the interest of the readers in testing the a-cubebenoate compound as an anti-cancer drug is high and presented in a straightforward way. However the biggest concern is the cell types that were used. It is ok to use murine cancer cell lines in a preliminary phase but the comparison with a normal human cell line is not really informative. It would be more informative to compare 2 murine cell lines or 2 human cell lines.  Also the "normal" cell line is not take along in comparison throughout the entire paper. It could very well be that other non cancerous cell lines exhibit the exact same thing which would mean the conclusion of the paper is entirely wrong. 

I strongly suggest to choose an appropriate set of cell lines to allow for appropriate comparisons and conclusions

Reviewer 4 Report

The title of manuscript “Anti-tumor effects of α-Cubebenoate derived from Schisandra chinensis in CT26 colon cancer cells” describes the content of study full. The data presented in the article are new and are of interest to researchers.

While reading, there were some remarks that can be corrected when editing the textю

  • Lines 40-41. The content of cubebenoate in the plant material should be added in order to have an idea of its availability
  • Line 43. DEPT, HMBC and HSQC are techniques of NMR. The sentence should be rewriting. Instead of “using various analytical chemistry techniques, which included polarization transfer (DEPT), heteronuclear multiple bond correlation (HMBC) nuclear magnetic resonance (NMR), and heteronuclear single quantum coherence (HSQC)” insert “using various NMR techniques, which included polarization transfer (DEPT), heteronuclear multiple bond correlation (HMBC) and heteronuclear single quantum coherence (HSQC)”
  • Line 66. Please explain how the doses of cubebenoate were chosen. Why are all doses lower than the cisplatin dose?
  • According to the structure shown in Fig. 1, cubebenoate has several chiral centers, but their stereochemistry is not indicated. What is known about whether a given substance is an individual compound, a mixture of diastereomers or enantiomers? What is the specific optical rotation?
  • Some paragraphs (lines 167-183) in discussion part should be rewriting in terms of SAR. Phenol compounds – cytotoxity and apoptosis, then gomisin - cytotoxity and apoptosis, then α-iso-cubebenol - cytotoxity and apoptosis. 320 µM – is not significant cytotoxicity.
  • The discussion part is poor due to lack of some results. But in results part there are some sentences discussed results. They should be moved from results to discussion, may be partially repeated. For example, "migration was significantly suppressed at α-cubebenoate concentrations of 30 µg/mL (Figure 3a), and a similar pattern was observed in the expression levels of key proteins of PI3K/AKT-mediated cell migration. Furthermore, α-cubebenoate decreased the levels of phosphorylated PI3K and AKT (Figure 3b). These results suggested that α-cubebenoate suppressed the migration of CT26 cells by regulating the PI3K/AKT signaling pathway".

And

"The phosphorylation levels of FAK and MLC were significantly decreased in LoCb, MiCb, and HiCb groups, although no dose-dependent response was observed (Figure 4b). These observations indicate that the inhibitory effects of α-cubebenoate on CT26 tumor growth might be associated with inhibition of adhesion ability via regulation of the FAK/MLC signaling pathway".

And other…

  • line 154. α-isocubebene